# Regional Distribution of Net Radiation over Different Ecohydrological Land Surfaces

**Md Masudur Rahman** [1,2,3], **Wanchang Zhang** [1,*] and **Arfan Arshad** [1,2]

[1]  Key Laboratory of Digital Earth Science, Aerospace Information Research Institute,
    Chinese Academy of Sciences, Beijing 100094, China; mmrahman@radi.ac.cn (M.M.R.);
    2012ag3729@uaf.edu.pk (A.A.)
[2]  University of Chinese Academy of Sciences, Beijing 100049, China
[3]  Department of Electrical and Electronic Engineering, Pabna University of Science & Technology,
    Pabna 6600, Bangladesh
*   Correspondence: zhangwc@radi.ac.cn; Tel.: +86-1810-129-8626

**Abstract:** Net radiation is an important component of the earth's surface energy balance, which plays a vital role in the evolution of regional climate or climate change. The estimation of this component at regional or global scales is critical and challenging due to the sparse and limited ground-based observations. This paper made an attempt to analyze the feasibility of a remote sensing-based surface energy balance model using satellite (TERRA/MODIS) data to derive the net radiation ($R_n$). In the present study, MODIS data at 15 different days of the year (DOY) were utilized to visualize the spatial pattern of net radiation flux over three versatile and heterogeneous ecohydrological land surfaces (upstream, midstream, and downstream) of northwest China (Heihe river basin). The results revealed that the estimated net radiation from the satellite data agrees well with the ground-based measurements over three different surfaces, with a mean relative error of 9.33% over the upstream superstation (grasslands), 13.95% over the middle stream superstation (croplands), and 11.63% over the downstream superstation (mixed forests), where the overall relative error was 11.64% with an overall *rmse* of 29.36 W/m$^2$ in the study area. The regional distribution of net radiation over the versatile land surfaces was validated well at a large scale during the five-month period and over different land surfaces. It was also observed that the spatial pattern of net radiation varies spatially over three different landscape regions during four different days of the year, which might be associated with different climatic conditions and landscape features in these regions. The overall findings of this study concluded that satellite-derived net radiation can rationally be obtained using a single-source remote sensing model over different land surfaces.

**Keywords:** net radiation; surface energy balance model; validation; different land surfaces; Heihe river basin of China

## 1. Introduction

Net radiation ($R_n$) is the balance between the total incoming solar radiations towards the Earth's surface and outgoing solar radiation fluxes, which mainly influences other surface energy balance components through evapotranspiration, photosynthesis, and heating of air and soil. Radiation fluxes on land surfaces are vital components of water and energy cycles due to their associated phenomena with different Earth systems, e.g., hydrosphere, atmosphere, and biosphere, which have made it a topic of great concern in several disciplines around the globe [1,2]. Therefore, accurate estimation of radiation fluxes on different ecohydrological land surfaces is necessary in surface energy balance [3–5]. Ground-based measurement networks provide accurate estimates of radiation fluxes; however, the lack

of sufficient measurement networks and measuring processes restricts the understanding of the changes in energy fluxes at the regional scale. Satellite observations along with ground-based measurements provide unique and effective processes to evaluate radiation fluxes at both regional and global scales. Recently, numerous studies have used several possible methods and models [6,7] for estimating net radiation along with other components based on satellite remote sensing data, namely: Surface Energy Balance Index (SEBI) [8], Simplified-SEBI (S-SEBI) [9], Surface Energy Balance System (SEBS) [10], Surface Energy Balance Algorithm for Land (SEBAL) [11], Mapping Evapotranspiration at High Resolution with Internalized Calibration) (METRIC) [12,13], Vegetation Index/Temperature Trapezoid (VIT) [14], and Two Source Model (TSM) [15]. Among these methods, the principle and model parameterizations, variations are involved in process-based source conditions (single or double), input requirements, assumptions, dependency on ground measurements and applications. These criteria demonstrated the degree of suitability of the methods over different surfaces at local and regional scales.

Nowadays, numerous satellites products, such as Landsat, Geostationary Operational Environmental Satellite (GOES), Advanced Very High Resolution Radiometer (AVRR), and Moderate Resolution Spectro Radiometer (MODIS), etc., are being utilized to estimate net radiation fluxes. Recently, apart from the large multifunctional Earth-observing satellites, small satellites have been dedicated to one particular observational task [16]. The advantages of this include simple and speedy design and manufacture, many more launch opportunities, low cost, non-competition between the requirements of different instruments, radiation fluxes assessment, risk reduction, etc. The MODIS onboard TERRA sensor [17] is an important instrument which provides the measurement of various land surface and atmospheric parameters with relatively high spatiotemporal resolution globally. Case studies conducted by [18,19] have proven that the results of $R_n$, derived using proposed schemes using MODIS from MODIS land and atmosphere data products, agree well with field measurements. The validation and estimation of satellite-derived surface fluxes are crucial and highly challenging, due to different land surface characteristics, which played a significant role in the estimation and validation processes [20]. In these circumstances, the present study has made an attempt to demonstrate the net radiation with the SEBS model using MODIS images over three different landscape regions of study areas such as the upstream, midstream, and downstream regions of the Heihe river basin in northwest China. The ground measurements of the surface energy balance components in the study region are sensitive due to complex topographic conditions, limitations in the installation of atmospheric instruments, as well as operation, maintenance and field campaigns being difficult to adapt [21–23]. However, ground-based measurements from three sets of high quality controlled 4-component net radiometers (4 CNR) were utilized for validation purposes, which are installed over three different landscape regions of the Heihe river basin in northwest China. The sample size of the validation work was relatively larger and comparison circumstances were distinctive by demonstrating some different ways as compared to earlier studies. The seasonal variations of estimated and measured net radiation were also compared during the vegetation growing period over the study area, and consequently, the spatial distribution of satellite-derived net radiation and land surface temperature was presented at the regional scale. The key purposes of this paper are two-fold: (i) to examine the regional distribution of net radiation ($R_n$) using TERRA-MODIS images over different ecohydrological land surfaces, and (ii) to validate the estimated $R_n$ with the 4-component net radiometer measurements over three superstations in the study area.

## 2. Materials and Methods

### 2.1. Study Area

The study area is located in the Heihe river basin of northwest China, which is geographically located between 97.24° E to 102.10° E and 37.41° N to 42.42° N, as shown in Figure 1. It is surrounded by the second largest inland river basin in China, approximately covering an area of 143,000 km$^2$.

The study region comprises different ecohydrological land surfaces from bare land to sparse vegetation, grassland and alpine grassland (upper stream), croplands (middle stream), and mixed forests (downstream). Average annual temperature varies over the study region between −40 °C (upper stream in winter) and 40 °C (downstream in summer). The maximum precipitation (>350 mm per year) is recorded over the upper stream, is about 100–250 mm per year over the middle stream, and the minimum precipitation is less than 50 mm per year over the downstream. The typical precipitation over the study area is approximately 110 mm per year. The elevation dynamics over the study area ranges between approximately 1000 (downstream) and 5000 m (upper stream).

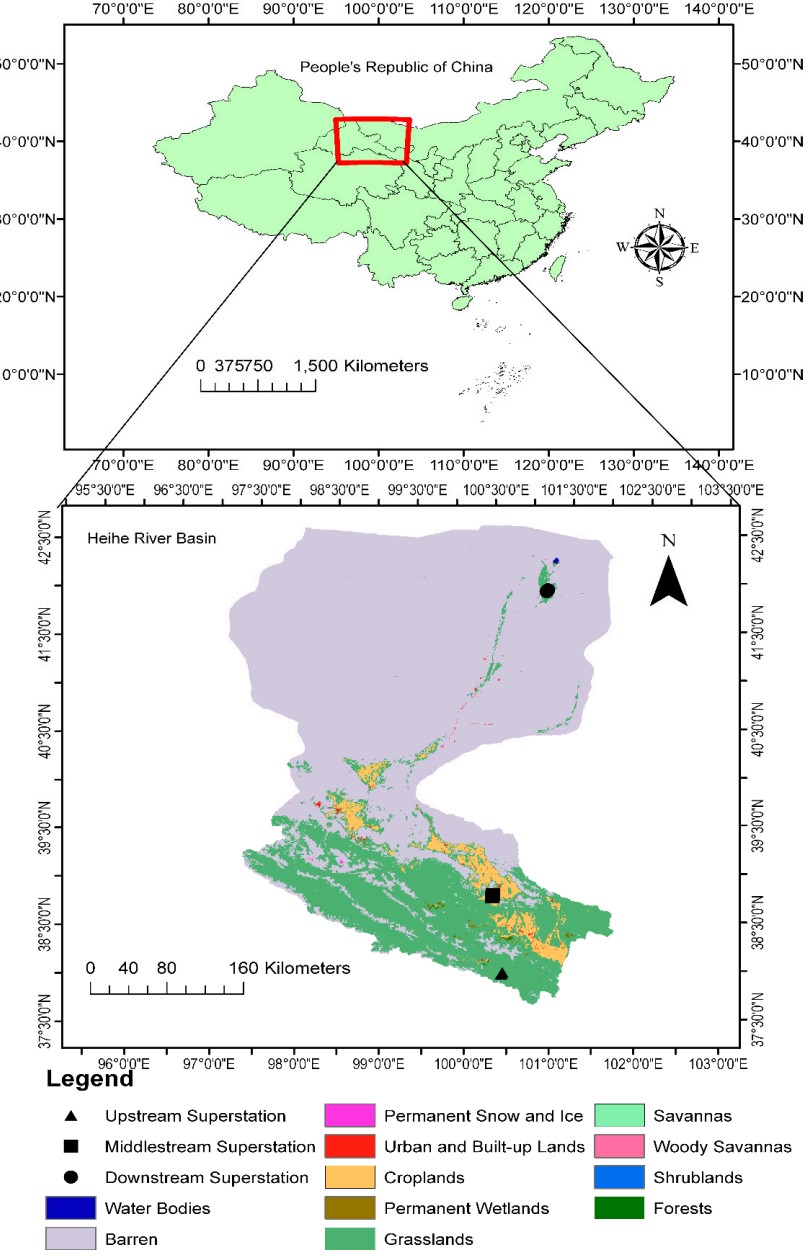

**Figure 1.** The study area with geographical coordinates and location of the measurement stations.

Several projects related to hydrological, environmental, energy and water exchange, and land surface processes have been carried out over this study area concerning the different ecohydrological land surfaces, such as the Heihe Basin Field Experiment (HEIFE) [24], Arid Environment Comprehensive Monitoring Plan 95 (AECMP' 95) [25], the Watershed Allied Telemetry Experimental Research project

(WATER) [26], and the Heihe WATER project (HiWATER) [27]. These multipurpose projects have facilitated a huge platform for the different important and crucial scientific experiments over the study area. The three superstations' (upstream, midstream, and downstream) 4-component net radiometer (4 CNR) data from the HiWATER project were utilized to perform the validation work of the MODIS-derived net radiation ($R_n$) i.e., the estimated $R_n$.

## 2.2. Ground Measurement and Instrumentation

Ground observation data of the net radiation ($R_n$) were taken from three (3) sets of 4-component net radiometer (4 CNR) instruments, which are installed over three superstations in different landscapes regions (up-, middle-, and downstream) to validate the regional distribution of the estimated net radiation ($R_n$). The measured net radiometer data were averaged to 30-min data. The experimental setup of the Heihe Watershed Allied Telemetry Experimental Research (HiWATER) project was utilized for the desired data collection. A detailed description about the geographical locations and the installation of these instruments is explained below.

The first set of the 4-component net radiometer (Kipp & Zonen, Delft, The Netherlands) is geographically located (100.464° E, 38.047° N) in Qilian (Qinghai province) at the upstream superstation. The land surface characteristics mainly comprise grassland and alpine grassland with an average height of 20–30 cm. The 4-component net radiometer was installed at 5 m height (south facing), where the altitude was of approximately 3033 m.

The second set of the 4-component net radiometer (Kipp & Zonen, Delft, and the Netherlands) is geographically located at 100.372° E, 38.855° N in Zhangye city of Gansu province over the middle stream superstation. The land surface characteristics comprise corn croplands, orchards, and greenhouses with an average height of 2–3 m. The 4-component net radiometer was installed at 10 m height (south facing), where the altitude was of about 1556 m.

The third set of the 4-component net radiometer (Kipp & Zonen, Delft, the Netherlands) with an installation height of 12 m (south facing) and altitude ~873 m is located at (101.137° E, 42.001° N) in the Ejina Banner of Inner Mongolia over the downstream superstation. The land surface characteristics vary, with mixed forests of Tamarix and Populus trees with an average height of 1 to 18 m.

## 2.3. Meteorological Data

Daily meteorological data of air temperature, relative humidity, wind speed, soil temperature, and soil moisture from May to September 2015 were utilized from meteorological observatories installed in study area, covering the Heihe River Basin. Those daily data of all the meteorological parameters used in this study were also obtained from the automatic weather station (AWS) of the Heihe Watershed Allied Telemetry Experimental Research (HiWATER) project over three streams (up, middle, and down).

## 2.4. Remote Sensing Data Preparation

Remote sensing data of calibrated and geolocated aperture radiances were taken from MODIS level 1B (MOD021KM) data and the corresponding geolocation data (MOD03) from May to September 2015. A total 15 images of MOD021KM with possibly clear-sky on different days of the year (DOY) were collected from the NASA-affiliated Level-1 and Atmosphere Archive and Distribution System website (https://ladsweb.modaps.eosdis.nasa.gov/). TERRA/MODIS images were utilized due to some benefits for our study purposes. Firstly, the study tools and the geographic information system-based remote sensing model (SEBS) were compatible with the MODIS images and their TOA (top of atmosphere) reflectance and radiance bands, as required for Rn estimation. Secondly, the spatial resolution of MODIS is optimum and regular for all bands but the spatial resolution of geostationary satellites is lower and irregular (band-dependent) for all bands such as 1 km for the visible band and 4 km for thermal bands of GEOS-8 and 9. Finally, the corresponding geolocation images (MOD03) of MODIS provide both the viewing azimuth and zenith angles for both the solar and sensor, respectively. These four angle maps

were used for the atmospheric corrections as they are essentially required for the surface energy balance system. The MRTSwath tool was used to re-project the raw data in HDF format to obtain a well-suited format for the geographic information system (GIS). The re-projected images were further processed to obtain the top of atmosphere reflectance (TOA) for bands 1 to 5 and 7 and radiance for bands 2 and 17 to 19, 31, and 32. However before using these bands in the SEBS model, atmospheric correction is essential, especially in surface energy balance issues. Therefore, the SMAC (simplified atmospheric correction) algorithm [28] was utilized to apply atmospheric corrections over the TOA reflectance bands. The aerosol optical thickness (AOT) data were obtained from https://aeronet.gsfc.nasa.gov/ and the ozone content was repossessed through https://ozoneaq.gsfc.nasa.gov/ for SMAC operation. We have utilized the data of the nearest AERONET (Aerosol Robotic Network) station (Dalanzadgad, northwest China) available on operation in our study area during the study period (May–September 2015). Due to the unavailability of all stations' data nearby our study area, we could not use any interpolation.

Atmospherically corrected reflectance and radiance bands were further used to acquire the key parameters to run the SEBS model. The most important parameter, surface albedo ($\alpha$), can be derived based on the following formula [29–31] as

$$\alpha = 0.160r_1 + 0.291r_2 + 0.243r_3 + 0.116r_4 + 0.112r_5 + 0.018r_7 - 0.0015 \tag{1}$$

where $r_{1\ to\ 7}$ indicates the corrected surface reflectance bands. In the experimental campaign, the near-ground albedo does not generally increase with increasing wavelengths for all kinds of surfaces. In the case of water surfaces, it was found that the albedo in the UV is more or less independent of the wavelength on a long-term basis. In the visible and near-infrared spectra, the water albedo obeys an almost constant power–law relationship with wavelength. In the case of sand surfaces, it was found that the sand albedo is a quadratic function of wavelength, which becomes more accurate if the UV wavelengths are neglected. Finally, the spectral dependence of snow albedo behaves similarly to that of water, i.e., both decrease from the UV to the near-IR wavelengths by 20–50%, despite the fact that their values differ by one order of magnitude [32].

Normalized Difference Vegetation Index (NDVI) was computed based on the visible red (VIR) band (band 1 of MODIS) and the near-infrared (NIR) band (band 2 of MODIS) [33] as

$$NDVI = \frac{NIR - VIR}{NIR + VIR} \tag{2}$$

The emissivity ($\epsilon$) over the different land surfaces was obtained by applying three basic conditions [34], which are being described as following

Condition 1: for bare soil pixels ($NDVI < 0.2$), the emissivity ($\varepsilon$) value was valued as

$$\epsilon = 0.9832 - 0.058VIR \tag{3}$$

Condition 2: for mixed pixels ($0.2 \leq NDVI \leq 0.5$), the emissivity ($\varepsilon$) value was computed as

$$\epsilon = 0.97 + 0.018f_c \tag{4}$$

Condition 3: for vegetation canopy pixels ($NDVI > 0.5$), the emissivity ($\varepsilon$) value was approximated as

$$\epsilon = 0.990 + \epsilon_d \tag{5}$$

where $f_c$ stands for vegetation propagation (fractional canopy coverage) and the $\epsilon_d$ stands for emissivity difference (which is 0.005 for the vegetation pixel). The $f_c$ was computed based on the empirical equation of the SEBS model using leaf area index (LAI) and NDVI as

$$LAI = \sqrt{NDVI\frac{1 + NDVI}{1 - NDVI}}$$

(6)

$$f_c = (\frac{NDVI - NDVI\_min}{NDVI_{max} - NDVI\_min})^2$$

(7)

The split window method was used to calculate the land surface temperature (*LST*) [33] as

$$LST = T_{31} + 1.02 + 1.79(T_{31} - T_{32}) + 1.2(T_{31} - T_{32})^2 + (34.83 - 0.68w)(1 - \varepsilon) + \\ (-0.73.27 - 5.19w)\epsilon_d$$

(8)

where $T_{31}$ and $T_{32}$ indicate the brightness temperature, acquired from the MODIS radiance bands 31 and 32, respectively, based on Planck's equation; the water vapor content ($w$) obtained from the MODIS radiance bands (17, 18, and 19); and $\epsilon_d$ is the emissivity difference.

## 2.5. SEBS Model Description

In this study, the surface energy balance system (SEBS) model developed by Professor Bob Su in the Netherlands [10] was utilized to estimate the net radiation flux from satellite data. The SEBS model has widely applied to estimate surface energy fluxes based on a single-source remote sensing (RS) model using available RS datasets. This model is more applicable and unique than other energy balance models due to some specific features. Firstly, two separate methods are used to calculate the surface fluxes: (i) the Monin–Obukhov similarity functions method is used if the reference height is below the top of the atmospheric surface layer (ASL); otherwise, (ii) the bulk atmospheric similarity method is used [35]. Secondly, the SEBS model computed the $KB^{-1}$ on a per pixel basis using an improved algorithm instead of other models (assuming the non-dimensional parameter $KB^{-1}$ as a constant). This feature combines a full cover canopy model [36], a bare land model [37], and a new scheme for vegetation–bare soil interaction [38,39]. This important feature facilitated our study area because it is combined of full cover canopy, vegetation bare land interaction, and bare land. Last but not least, two limiting conditions are considered: dry and wet limits (the latent heat flux is zero at the dry limit, i.e., the sensible heat flux ($H_{dry}$) is maximum (the available energy, $R_n - G$); at the wet limit, $H_{wet}$ is determined using the reversed Penman–Monteith equation in terms of evaporative fraction ($EF_r$) for every pixel [10,40].

In the surface energy balance system (SEBS), the equation of net radiation ($R_n$) can be written as

$$R_n = G_0 + H + \lambda E$$

(9)

where $G_0$ is the soil heat flux, $H$ is the sensible heat flux, and $\lambda E$ is the latent heat flux (H and LE are the turbulent fluxes).

Under the given meteorological conditions, the friction velocity ($u_*$) and sensible heat flux ($H$) are computed at the reference height on the basis of similarity relationships as

$$u_* = uK[\ln\left(\frac{z - d_0}{z_{0m}}\right) - \psi_m\left(\frac{z - d_0}{L}\right) + \psi_m\left(\frac{z_{0m}}{L}\right)]^{-1}$$

(10)

$$H = ku_*\rho C_p(\theta_0 - \theta_a)[\ln\left(\frac{z - d}{z_{0m}}\right) - \psi_h\left(\frac{z - d}{L}\right) + \psi_h\left(\frac{z_{0h}}{L}\right)]^{-1}$$

(11)

where $K$ is the von Kármán constant; $u_*$ is the friction velocity; $z$ is the measurement height above the surface; $d_0$ is the zero-plane displacement height; $z_{0m}$ and $z_{0h}$ are the roughness height for momentum

and heat transfer, respectively; $\psi_m$ and $\psi_h$ are the stability correction functions for momentum and sensible heat transfer, respectively. $L$ is the Obukhov length, which can be defined as

$$L = -\frac{\rho C_p u_*^3 \theta_v}{K g H} \tag{12}$$

where $\rho$ is the density of air; $g$ is the gravitational acceleration (ms$^{-2}$); $C_p$ is the specific heat for air at constant pressure; $\theta_v$ is the potential virtual temperature at near surface (K).

The above three nonlinear equations (10)–(12) can be solved on the basis of the Broyden method [10]. Essentially, the calculation of *H*, as described above, is independent of other surface energy balance components because it requires the land surface temperature (*LST*) and meteorological variables. If there are uncertainties in the *LST* and meteorological measurements, the estimated value of H can be affected. To minimize this uncertainty, the SEBS model considers the energy balance of limiting cases, as the value of H should fall between the dry and wet limits (extreme limits) [10,40].

At the dry limit condition, the $\lambda E$ becomes the minimum value ($\lambda E \approx 0$) and the $H_{dry}$ can be computed as

$$H_{dry} = R_n - G_0 \tag{13}$$

At the wet limit condition, the *H* becomes minimum and the $\lambda E$ could be maximum and the $H_{wet}$ can be computed as

$$H_{wet} = \frac{R_n - G_0 - \left(\frac{\rho C_p}{r_{ew}}\right).(e_s - e)/\gamma}{1 + \left(\frac{\Delta}{\gamma}\right)} \tag{14}$$

where $R_n$ is the net radiation, $r_{ew}$ is the external (aerodynamic) resistance (s/m), $e$ and $e_s$ are the actual and saturation vapor pressure (*Pa*), respectively, $\Delta$ is the rate of change of the saturation vapor pressure (*Pa/K*), and $G_0$ is the soil heat flux (W/m$^2$). The latent heat flux ($\lambda E$) can be computed in terms of evaporative fraction (*EF*) as

$$\lambda E = EF(R_n - G_0) \tag{15}$$

$$EF = \frac{EF_r * \lambda E_{wet}}{R_n - G_0} \tag{16}$$

$$EF_r = \frac{H_{dry} - H}{H_{dry} - H_{wet}} \tag{17}$$

where *EF* is the evaporative fraction and $EF_r$ is the relative evaporative fraction; for more details, please see [10,41].

The balance of solar radiation (incoming and outgoing) is the net radiation ($R_n$) as

$$R_n = [(1 - \alpha)R_{swd}] + \left[\varepsilon \varepsilon_a \sigma \theta_0^4 - \varepsilon \sigma \theta_a^4\right] \tag{18}$$

where $R_{swd}$ is the downward shortwave solar radiation (W/m$^2$), the dimensionless parameters $\varepsilon$ and $\varepsilon_a$ are the surface and atmospheric emissivity, respectively, $\theta_0$ is the potential surface temperature, $\theta_a$ is the potential air temperature, and $\sigma$ is the Stefan–Boltzmann constant (5.670373*10$^{-8}$ Wm$^{-2}$K$^{-4}$).

The soil heat flux ($G_0$) can be calculated as

$$G_0 = R_n[\Gamma_c + (1 - f_c)(\Gamma_s - \Gamma_c)] \tag{19}$$

Here, the ratio of soil heat flux to net radiation is assumed as $\Gamma_c$ = 0.05 for full vegetation canopy [42] and $\Gamma_s$ = 0.315 for bare soil [43]. An interpolation is then performed between these limiting cases using the fractional canopy coverage, $f_c$, which can be determined from remote sensing data (please see Equations (6) and (7)). For more detailed information on SEBS model parametrization, readers are directed to [10,44].

Two types of inputs are needed to run the SEBS model: (1) remote sensing image-based physical parameters such as land surface temperature (LST), albedo ($\alpha$), emissivity ($\varepsilon$), normalized difference vegetation index (NDVI), leaf area index (LAI), and fractional vegetation cover ($f_c$); (2) the weather parameters at the reference height of field-based instrumental measurements such as downward solar radiation, wind speed, air temperature, specific humidity, and air pressure. In this study, the ground-based parameters were obtained from the NOAH land surface model of GLDAS 2.1 (Global Land Data Assimilation System), where the spatiotemporal resolution was of 0.25°, resampled to 1 km and of 3 h, respectively. The main outputs of the SEBS model with these input parameters are net radiation ($R_n$), soil heat flux ($G_0$), sensible heat flux ($H$), and latent heat flux ($\lambda E$). In the results analysis of this study, we have examined the net radiation ($R_n$).

### 2.6. Statistical Evaluation

The above said SEBS model was applied to estimate net radiation (Rn) using 15 MODIS clear-sky images from May to September 2015 over the study region. The estimated net radiation (Rn) from remote sensing data was compared with ground-based net radiation (Rn) from 4-component net radiometer measurements over three superstations. The performance and feasibility of the SEBS model in predicting net radiation (Rn) was diagnosed based on three statistical indicators: the error (bias), relative error (re), and the rmse using the following equations [45].

$$re = \frac{R_{n(estimated)} - R_{n(measured)}}{R_{n(measured)}} * 100\% \tag{20}$$

$$rmse = \sqrt{\frac{\sum_{i=1}^{n} R_{n(estimated)} - R_{n(measured)}}{N}} \tag{21}$$

where $R_{n(measured)}$ is the ground-based measured value from the 4-component net radiometer and $R_{n(estimated)}$ is the satellite-estimated value from MODIS images. The case study and validation, monthly variations, and spatial pattern illustration of the net radiation estimation over heterogeneous landscapes have been presented and will be explained in the next sections.

## 3. Results and Discussions

### 3.1. Hydrometeorological Conditions over Study Region

Figure 2 illustrates the daily time series variation in hydroclimatic conditions from May to December 2015 over the different landscape regions of the study area. Overall daily air temperature over all landscape regions increases from May and reaches a maximum peak in August and then, starts to decrease until December; however, the value of maximum air temperature was higher at ~30 °C in the downstream region as compared to the midstream and upstream regions at 25 and 15 °C, respectively. There are very large differences (approximately from 1000 m (downstream) to 5000 m (upstream)) in terms of altitude for these three stations. The temperature differences are certainly linked to their respective altitudes as well. Relative humidity over all landscape regions varies within 10–60% over the downstream region, 20–90% over the midstream region, and 40–90% over the upstream region in a fluctuating trend. The lower values of relative humidity over the downstream region correspond to higher temperatures values. Daily time series variations in soil temperatures followed the pattern of air temperature, where it increases from May and reaches to a peak value in August and then, starts to decrease until December. Soil temperatures over the upstream, midstream, and downstream regions vary between −5 and 10 °C, −5 and 23 °C, and −3 and 24 °C, respectively. Higher values of soil temperature in the downstream regions might be associated with higher air temperature, sparse vegetation, and barren land. Daily soil moisture continuously decreases from May to December; however, the minimum (maximum) values of soil moisture could be seen as 5% (29%), 20% (40%), and 29% (39%) over the upstream, midstream, and downstream regions, respectively.

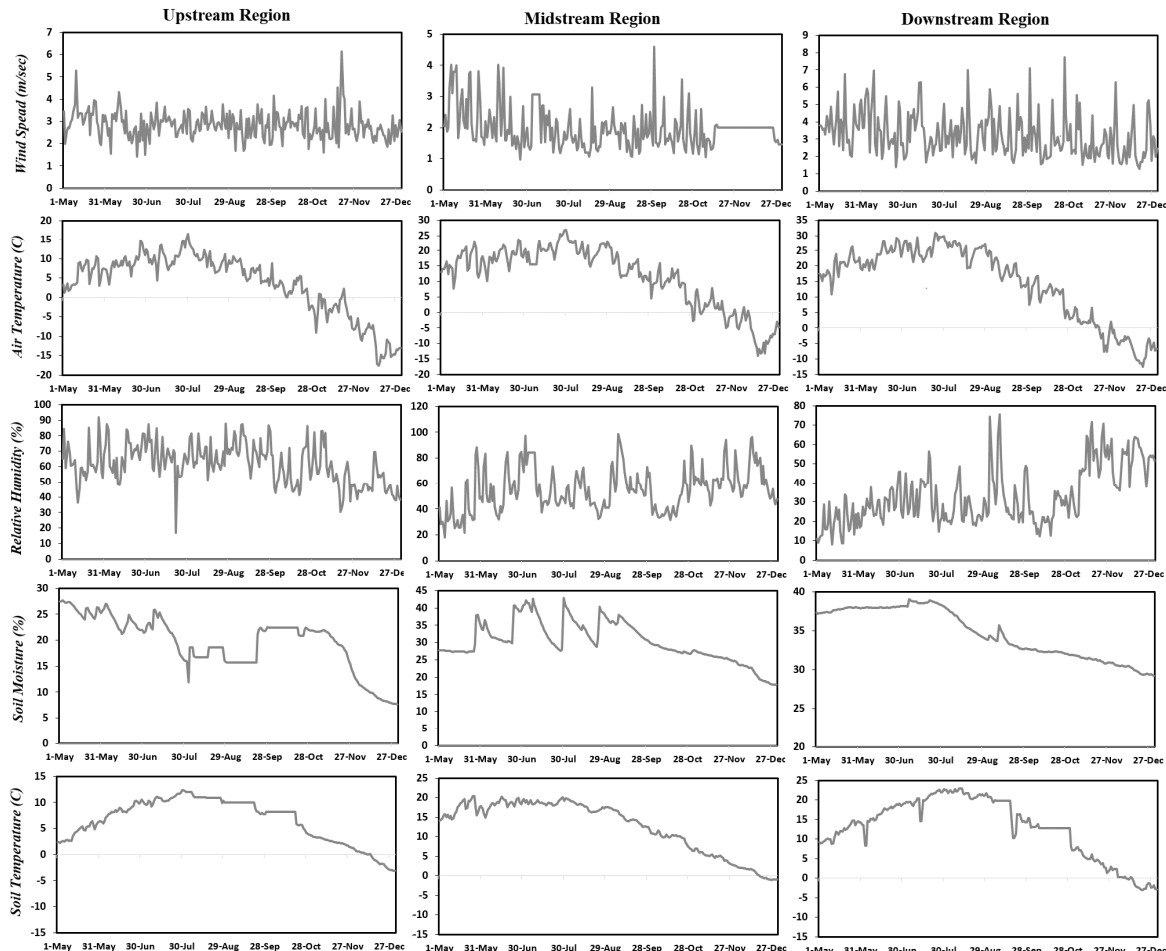

**Figure 2.** Daily time series plots of hydrometeorological parameters: wind speed, air temperature, relative humidity, soil temperature, and soil moisture from May to December 2015 over three different landscape regions.

## 3.2. Estimation of Net Radiation ($R_n$) and Ground Validation

The time resolution of the measured Rn data used in the validation was half an hour (30 min) averaged near the satellite over the passing moment. The estimated $R_n$ obtained from the satellite-based SEBS model for different days of the year (DOYs) only around the three different superstations (up-, middle-, and downstream) were compared with the respective measured $R_n$ data. The calculated statistical parameters were the coefficients of determination ($R^2$), root mean square error (rmse), and bias. Figure 3 illustrates the scatter plot sketched between measured and estimated $R_n$ values at three measurement stations over three different landscape regions in the study area. The linear regression analysis result showed that the slope of the regression was very close to unity with the optimum correlation coefficient, as shown in Figure 3, where the overall *rmse* was of 29.36 W/m². Higher values of coefficient of determination ($R^2$) over upstream, midstream, and downstream regions were found as 0.99, 0.97, and 0.98, respectively, which indicate the estimated $R_n$ obtained from proposed approach correlate better with ground measurements.

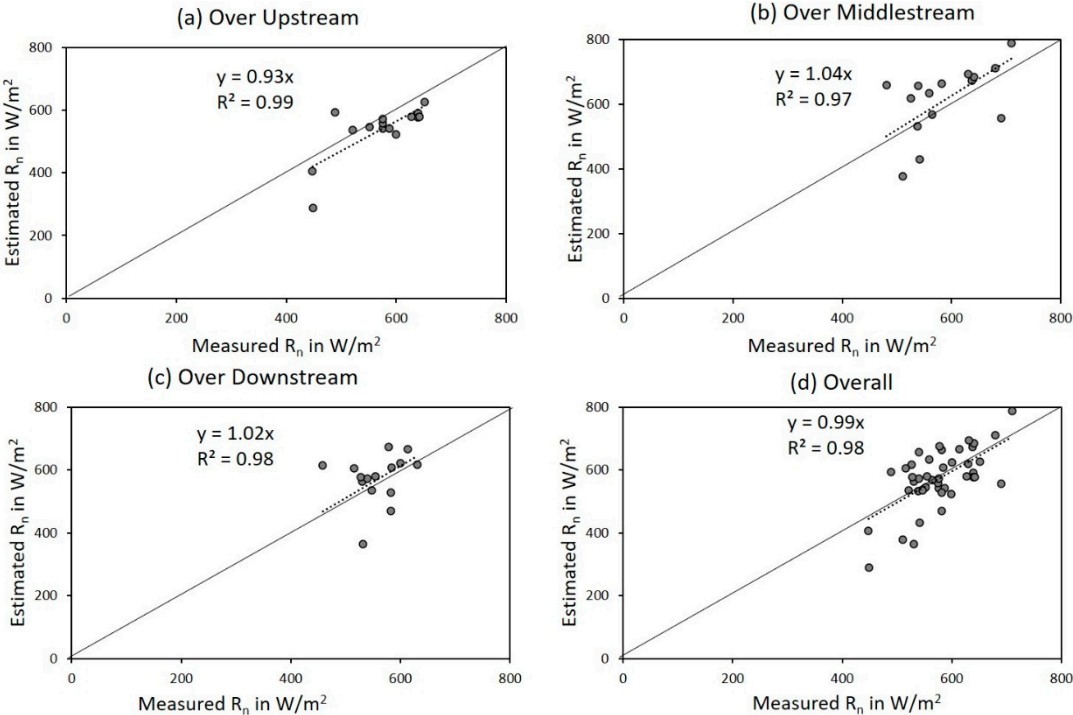

**Figure 3.** Comparison of satellite estimated net radiation with field measurements over (**a**) upstream, (**b**) midstream, (**c**) downstream, and (**d**) overall averaged over three landscape regions.

Table 1 summarized the detail description of statistical parameters evaluated to validate the estimated net radiation ($R_n$) over three different landscape regions. It was seen that net radiation ($R_n$) derived from remote sensing data underestimated the ground measurements over the upstream and overestimated over the midstream and downstream regions. Results indicate that the average value of error (bias) over upstream, midstream, and downstream regions was found as −34.54, 27.82, and 14.45, respectively. The overall mean relative error (*re*) of 11.64%, with separately as 9.33% over upstream (underlying surface is alpine grasslands), 13.95% over middle stream (underlying surface is croplands), and 11.63% over downstream (underlying surface is mixed forests) superstations, respectively. The computed rmse was of 10.98 W/m² over upstream (grassland), 34.04 W/m² over middlestream (cropland), and 43.05 W/m² over downstream (mixed forest), with an average *rmse* of 29.36 W/m². Over and under estimation in net radiations might be due to fact that it is estimated by using the vegetation index, which is sensitive to solar positions and changes in soil reflectance [46]. Moreover, net radiation flux is also sensitive to land surface temperature [47]; however, land surface temperature estimated from MODIS shows higher uncertainty than air temperature, which definitely leads to underestimation in net radiations [45]. The overall relative error (re) is relatively higher because the satellite-based estimation of energy fluxes is performed in fairly diverse and unconnected ways. As per most of the previous studies in this research community [48], it was concluded that the relative error (*re*) in the satellite estimation of energy fluxes is expected to be found up to 30% and the average *rmse* is expected to be found up to 50 W/m². Therefore, the satellite estimation of $R_n$ agrees well with the ground measurements over this study area. The estimation of net radiation ($R_n$) using a single-source remote sensing model is satisfactory, and it shows better performance than the estimation of sensible heat flux (H) over different landscape regions in northwest China [20].

**Table 1.** The error (bias) and relative error (re) of the Rn estimation with respect to the field measurements.

| Day of Year | Upstream | | Middlestream | | Downstream | |
|---|---|---|---|---|---|---|
| | Bias (W/m$^2$) | re (%) | Bias (W/m$^2$) | re (%) | Bias (W/m$^2$) | re (%) |
| 124 | −42.52 | 9.50 | −131.8 | 25.84 | −166.7 | 31.39 |
| 138 | −75.72 | 12.64 | −6.00 | 1.11 | 23.62 | 4.04 |
| 145 | −47.87 | 7.49 | 3.69 | 0.65 | 33.42 | 6.30 |
| 152 | −160.2 | 35.72 | −110.9 | 20.48 | −54.98 | 9.44 |
| 161 | −63.1 | 9.87 | 35.55 | 5.58 | 89.68 | 17.39 |
| 177 | −35.08 | 6.09 | −133.9 | 19.39 | −111.8 | 19.22 |
| 190 | −26.49 | 4.07 | 30.99 | 4.56 | −12.41 | 1.97 |
| 205 | −47.94 | 7.64 | 43.74 | 6.82 | 51.35 | 8.37 |
| 209 | −6.83 | 1.24 | 77.92 | 10.97 | 95.72 | 16.55 |
| 216 | −64.34 | 10.02 | 62.06 | 9.83 | 22.38 | 3.73 |
| 229 | −18.2 | 3.16 | 91.66 | 17.42 | 32.65 | 6.05 |
| 234 | −45.53 | 7.75 | 82.44 | 14.16 | 24.36 | 4.39 |
| 248 | −3.37 | 0.58 | 117.19 | 21.72 | −13.03 | 2.38 |
| 257 | 14.7 | 2.82 | 75.40 | 13.50 | 48.85 | 9.26 |
| 268 | 104.33 | 21.35 | 179.36 | 37.31 | 155.85 | 34.01 |
| Average | −34.54 | 9.33 | 27.82 | 13.95 | 14.59 | 11.63 |

Figure 4 illustrates the frequency distribution of $R_n$ on different days of the year over the study region. The mean values of the estimated $R_n$ were 369.18 W/m$^2$ on 4 May, 609.43 W/m$^2$ on 28 July, 549.50 W/m$^2$ on 4 August, and 556.56 W/m$^2$ on 5 September, as shown in Figure 4. Seasonally, the $R_n$ increased from May to August and then, decreased until September as the seasons changed from spring to autumn. Overall, the frequency distribution of net radiation is lower in the pre-monsoon period (May) and higher at the beginning of the monsoon (July) due to clear skies and then, decreased again in the post-monsoon period (September). The similar changing trend was also explained by a potential study over the Himalayan slope [49].

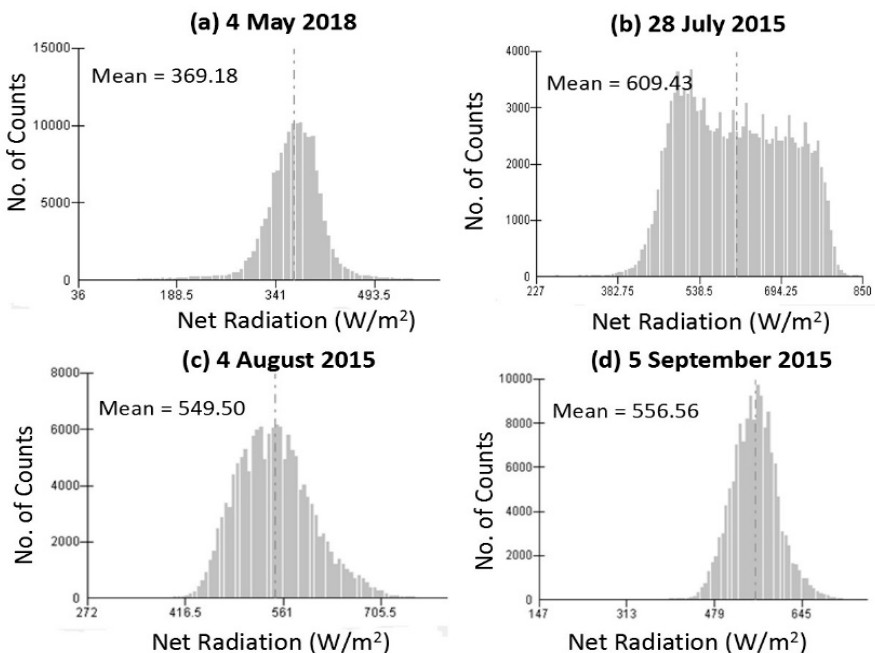

**Figure 4.** The $R_n$ frequency distribution over the study area on the selected four days as: (**a**) on 4 May, (**b**) on 28 July, (**c**) on 4 August, and (**d**) on 5 September of 2015.

Comparisons of Monthly Variations in $R_n$ Estimation over Different Landscapes

In Figure 5, we have used the 15 instant estimated values of $R_n$ during 15 DOYs of the period May–September 2015 over three different stations, where each month contains three data points, and the respective measured values of $R_n$ from ground net radiometer. Here, we want to observe the monthly changes in the instant values of the estimated and the measured $R_n$ for the study period. Figure 5 illustrated the comparison between monthly remote sensing-based $R_n$ obtained from the SEBS model and the ground-based measurements, specifically over three different landscape regions. It was seen that the time series of monthly $R_n$ estimated from the SEBS model shows a similar pattern to ground-based measurements over the three landscape regions; however, for the upstream region, estimated $R_n$ values were much closer to ground-based measurements than the other regions. Monthly variations in $R_n$ values indicated that it is obviously lower in May and June, which may be associated with heavy rain and clouds in monsoon seasons, while it reaches peaks in July–August and then, starts to reduce until December. A comparatively similar variation was also found in some previous potential studies [20,50]. The fluctuating trend in $R_n$ values was similar over three different land surfaces; however, the maximum value of $R_n$ (~800 w/m$^2$ in August) was higher in the midstream region as compared to other regions. Declining trends were estimated from June to July, and growing trends from August to September, but the variation was not so noticeable. The average value of $R_n$ over the three different lands surfaces was also similar in monthly variations, i.e., the use of a single-source RS model like SEBS is suitable for different types of land surfaces.

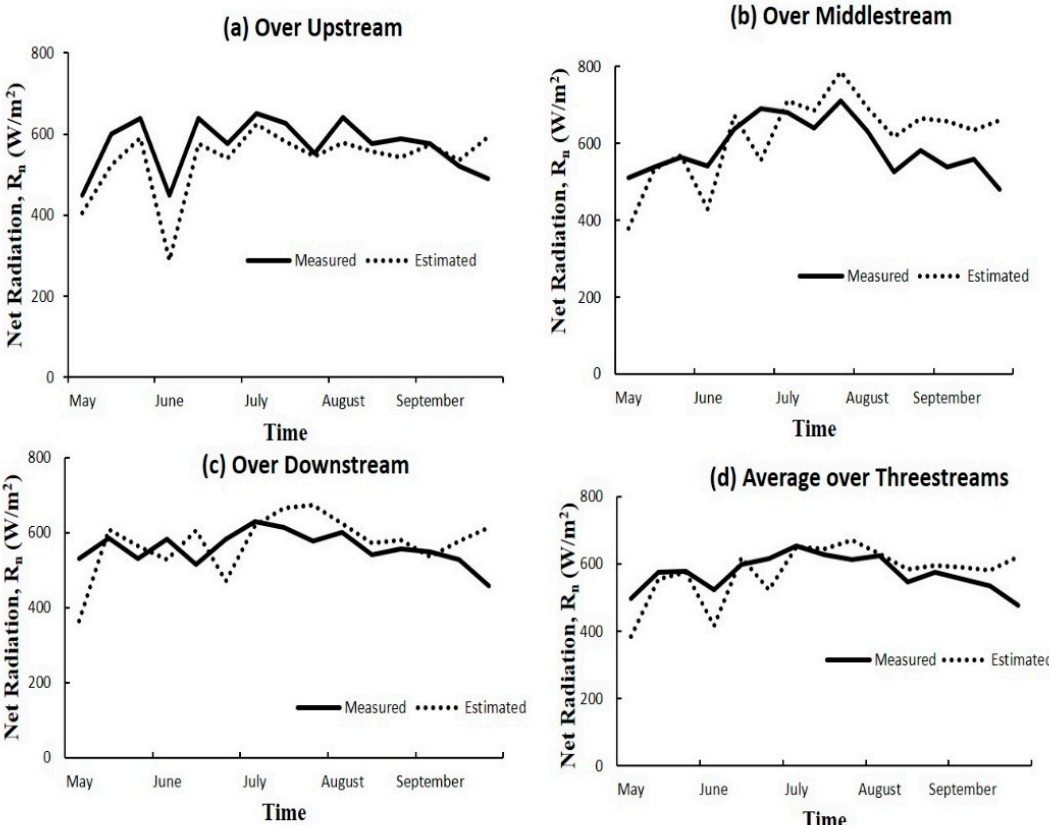

**Figure 5.** The seasonal variations of $R_n$ estimated from MODIS scenes with the SEBS model and measured by net radiometers over three different land covers as: (**a**) over upstream, (**b**) over middle stream, (**c**) over downstream (**c**), and (**d**) average over three streams.

### 3.3. Spatial Distribution Pattern of Rn

Figure 6 illustrates the regional distribution pattern of $R_n$ over the study area on different days of the year as 4 May, 28 July, 4 August, and 5 September of 2015. Its value ranged from 36 to 646 W/m$^2$ on 4 May, 272 to 850 W/m$^2$ on 28 July, 227 to 850 W/m$^2$ on 4 August, and 147 to 811 W/m$^2$ on 5 September. It was seen that the spatial pattern of net radiation varies spatially during four different days of the year, which may be associated with changes in landscape features and meteorological conditions over the study regions. Spatial variations in $R_n$ illustrated that its values are lower in midstream regions and higher in upstream and downstream regions over the study area. The higher values of $R_n$ in the downstream region might be associated with higher soil temperature and lower values of soil moisture existing in that region. It was seen that in the downstream region, the average $R_n$ values increased from ~240 W/m$^2$ (in May) to 827 W/m$^2$ (in July) and then, decreased to 560 W/m$^2$ (in August) and 398 W/m$^2$ (in September), while a reverse phenomenon could be observed in the downstream region, where $R_n$ values were lower from May to July and reached a maximum value (~840 W/m$^2$) in August–September, which might be due to different seasonal climatic changes and landscape features in both regions. The spatial distribution pattern of $R_n$ was also compared with the *LST* on the same days of the year over the study area (Figure 7). It was observed that spatial changes in $R_n$ are closely associated with the spatial pattern of LST because it is a key factor for estimation of surface fluxes. A similar phenomenon was also observed in the previous studies [20,50]. Some parts of Figures 6b and 7b were contaminated with clouds because the rainy season prevailed during the image collecting time over the study area.

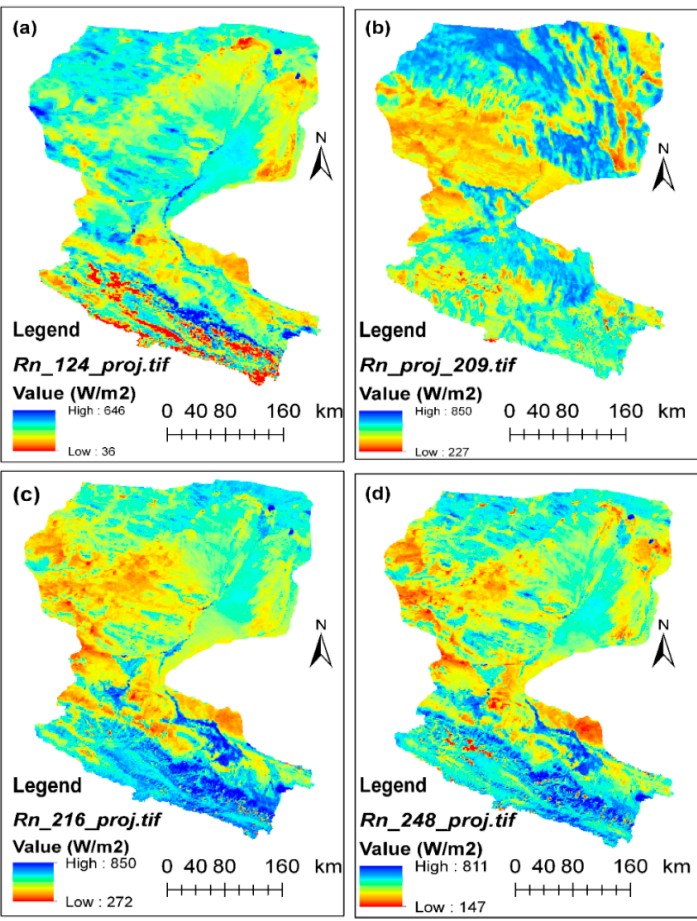

**Figure 6.** The regional pattern distribution of $R_n$ (W/m$^2$) over the study area as: (**a**) on 4 May, (**b**) on 28 July, (**c**) on 4 August, and (**d**) on 5 September of 2015.

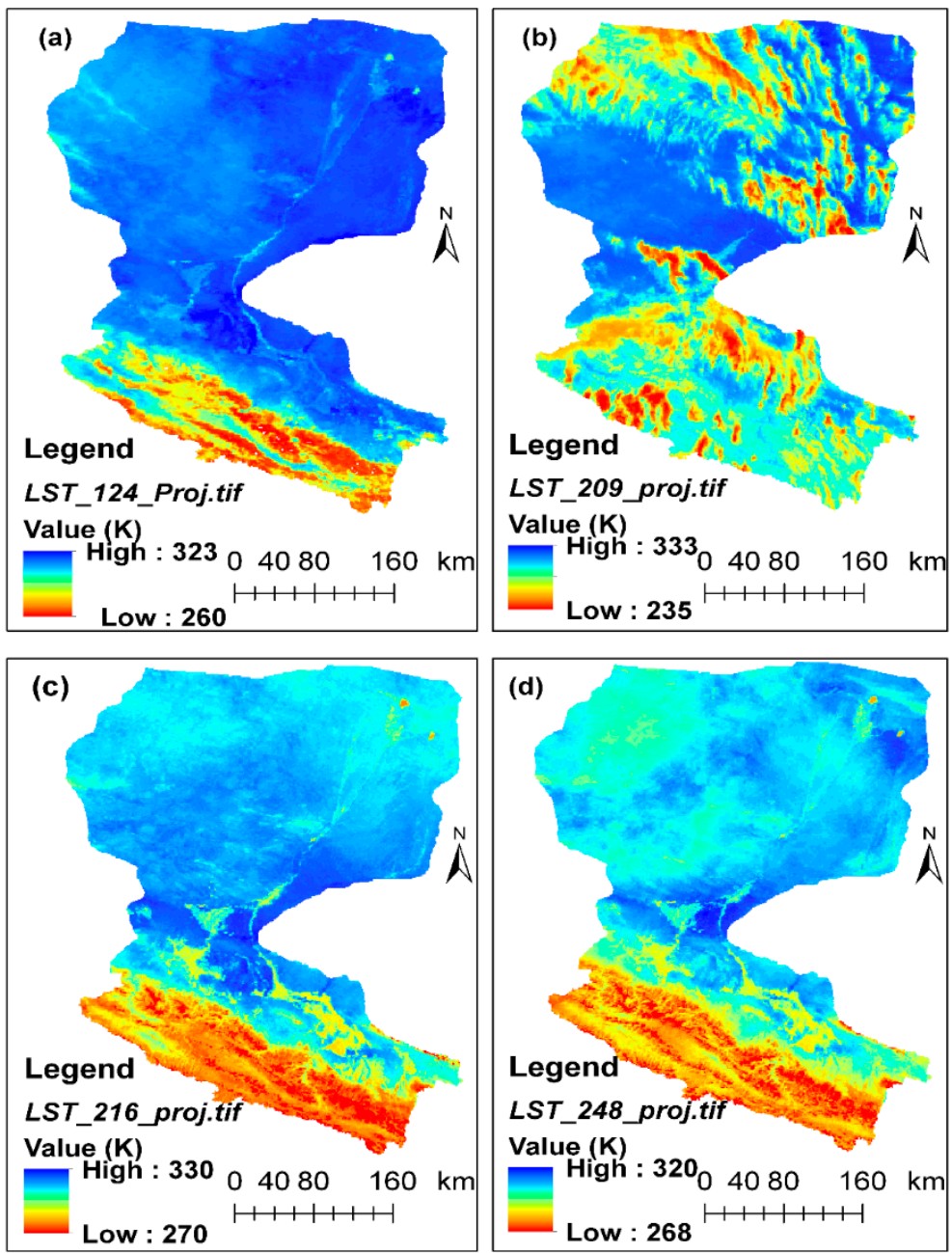

**Figure 7.** The regional pattern distribution of *LST* (K) over the study area as: (**a**) on 4 May, (**b**) on 28 July, (**c**) on 4 August, and (**d**) on 5 September of 2015.

## 4. Conclusions

Mainly, this study tested the viability of net radiation estimation using a single-source remote sensing model and satellite (MODIS) data over different ecohydrological surfaces in northwest China. The public availability of MODIS images and the suitable processing in single-source models like SEBS motivated this study. Comparison of our estimated results with ground measurements over three different land surfaces (alpine grassland, cropland, and mixed forest) provides overall correlation coefficients ($R^2$) of 0.98, with overall rmse of 29.36 W/m$^2$ (up to 50 W/m$^2$ is expected to be found in satellite estimation), which suggests that this study is a good approach for estimating net radiation over different land surfaces. The large-scale validation outcomes exposed that the estimation was performed well in the study area, but there was noticeable overestimation and underestimation over different superstations.

To understand the phenomena of these over and underestimations, one should address the following facts: (i) the net radiometer that measured $R_n$ was a point-based technique, but the estimation of $R_n$ used MODIS images with the SEBS model and GLDAS model, where values were averaged over a wide and larger area; (ii) the radiometer-based measurement was a half-hour averaged value, but the satellite estimated value was calculated by instantaneous observations of the surface parameters and thermodynamic states during the satellite over passing time; (iii) the transformation of the bare land to full vegetation cover during the transition/growing period (May to September was also the growing period in this study area) caused the complicated thermodynamic state; (iv) the surface energy balance was influenced due to surface and vegetation heterogeneity; (v) in the case of ground and wind profile measurements, there may have been missing values in the validation of roughness length estimation; and (vi) the calculation of land surface temperature (the key parameter in the computation of surface energy balance components) can be affected by the approximation in the calculation of the surface emissivity. The estimation of $R_n$ in this study is also affected by these factors, which are indicated as the causes for the over and underestimations.

The regional distribution pattern of $R_n$ presented an intermediate contrast due to the different land surface characteristics. The current approach used in this study is feasible to estimate the $R_n$ over a regional scale using the publicly available MODIS data and simplicity in data analysis. However, certain limitations still exist in the present methods, which may be subjected to future investigation. Calibration of model parameters is a necessary step to investigate the uncertainties in model predictions, while the current study does not compare the estimated and ground-based downward solar radiation and surface albedo due to the lack of measurement data in this study region during the study period. Therefore, a future study on the comparison and validation of downward solar radiation and albedo might explain partially the uncertainties in estimating net radiation ($R_n$) over different land surfaces.

**Author Contributions:** The first author M.M.R. designed this study. M.M.R. performed the data collection, model derivation, and validation. The weather data were analyzed by A.A. The second author W.Z. was the supervisor of this work and contributed with continuous guidance during this work. M.M.R. wrote this manuscript and it was edited by W.Z. and A.A. All authors have read and agreed to the published version of the manuscript.

**Funding:** This work was supported in part by the National Key Research & Development Program of China (grant numbers 2016YFA0602302 and 2016YFB0502502).

**Acknowledgments:** The first author is grateful to the authority of Pabna University of Science and Technology, Bangladesh for providing the intellectual contribution. This study also wishes to acknowledge the intellectual and material contributions of CAS-TWAS President's Fellowship Award. The authors are grateful to the anonymous reviewers for their constructive comments and suggestions to improve this manuscript.

**Conflicts of Interest:** The authors declare no conflict of interest.

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
