# Peer review of "Regional Distribution of Net Radiation over Different Ecohydrological Land Surfaces"

_atmosphere, doi:10.3390/atmos11111229_

Round 1

Reviewer 1 Report

In this manuscript, based on the SEBS model, the authors use MODIS and other meteorological data to calculate the net radiative flux over three versatile and heterogeneous land surfaces of northwest China. The validation against ground observation data shows overall RBE of 11.64% and RMSE of 29.36 W/m2 in the study area. The study is a good contribution to the community and is in line with the scope of the journal. However, there are still several points that need to be clarified. I recommend a major revision before publication.

Major points:

  • What is the time resolution of the Rn data used in the validation? Are they the daily average or the half-hour averaged data near the satellite over-passing moment? If it is the daily mean value, could the authors precise how the extrapolation is done for the hours before and after the satellite observation time?
  • The observation frequency of MODIS observation is relatively low, as the author found in total 15 clear-sky cases during the study period. Why not use geostationary satellite instead? There will be better spatial and temporal coverage, though spatial resolution is lower. I have doubt that with such few cases of observation/retrieval, this will be very useful for the monitoring of net surface radiative flux on a regular basis. Could the author give more detailed suggestions for the benefits of these data?
  • Overall, 10%+ bias is not small. This means the evaluation of net radiative flux is still very difficult. I think this is a point should be acknowledged by the authors.
  • P4, L151-152: are the aerosol data from AERONET collocated with the radiation observation stations? How these data are used? Any interpolations?
  • Figure 5: I doubt that these 15 values calculated in this study can be used as ‘monthly values’. They are 15 instants of clear-sky values and they don’t strictly represent monthly changes. This figure appears to me very confusing.

Minor points:

  • P3, L93: a space is needed before ‘143000’.
  • P6, L196 & L199: a pace is need after ‘where’.
  • P7, L230-231: there are very large differences in terms of altitude for these three stations. The temperature differences are certainly linked to their respective altitudes as well.
  • Table 1: I suggest to change the caption from ‘Error (Bias)’ to ‘Bias (W/m2)’.
  • P11, L321: ~240 W/m2, how this value is calculated? Average value? Same question for the other three values in July, August and September.
  • Figure 6: I would like to see the original MODIS images (composite, VIS or similar) for these four cases as well. Is there any cloud in each region (small cumulus clouds or thin cirrus cloud)?

Author Response

Dear Reviewer,

Thank you very much for your comments concerning our manuscript (atmosphere-964561). These comments are valuable and very much helpful for improving our paper. We have studied your comments carefully and have revised our manuscript accordingly, which we hope meet with your approval. The changed and added portion have been highlighted in the revised manuscript with the changes tracked.

The point by point responses are given in the attachment, Please see the attachment.

Best Regards.

Sincerely yours,

Dr. Md Masudur Rahman

Dr. Wanchang Zhang

Arfan Arshad

Reviewer 2 Report

The paper focuses on an interesting topic. The results obtained are of interest to the remote sensing society and the associated topics. I do believe that the paper merits publication after taking into account the following revisions:

1) Recently, apart from the large multifunctional Earth‐observing satellites, to small satellites dedicated to one particular observational task. The advantages of this include simple and speedy design and manufacture, many more launch opportunities, low cost, non‐competition between the requirements of different instruments, radiation fluxes assessment, risk reduction, etc. That is why a very brief reference on this is missing from the introduction (e.g. https://www.tandfonline.com/doi/full/10.1080/01431160701347147).

2) The results obtained from experimental campaigns show that the near-ground albedo does not generally increase with increasing wavelengths for all kinds of surfaces. In the case of water surfaces it was found that the albedo in the UV is more or less independent of the wavelength on a long-term basis. In
the visible and near-infrared spectra the water albedo obeys an almost constant power-law relationship with wavelength. In the case of sand surfaces it was found that the sand albedo is a quadratic function of wavelength, which becomes more accurate if the UV wavelengths are neglected. Finally, the spectral dependence of snow albedo behaves similarly to that of water, i.e. both decrease from the UV to the near-IR wavelengths by 20–50 %, despite the fact that their values differ by one order of magnitude (water albedo being lower). This kind of information is missing from the discussion on the albedo and therefore it should be added.( e.g. https://core.ac.uk/download/pdf/193528355.pdf)

Author Response

Dear Reviewer,

Thank you very much for your comments concerning our manuscript (atmosphere-964561). These comments are valuable and very much helpful for improving our paper. We have studied your comments carefully and have revised our manuscript accordingly, which we hope meet with your approval. The changed and added portion have been highlighted in the revised manuscript with the changes tracked. The point by point responses are given in the attachment, please see the attachment.

Best Regards.

Sincerely yours,

Dr. Md Masudur Rahman

Dr. Wanchang Zhang

Arfan Arshad

Reviewer 3 Report

Review for “Regional distribution of net radiation flux of surface energy balance over different ecohydrological land surfaces”

Summary: The manuscript compared the estimated net radiation from MODIS images with ground-measured net radiation at three sites in Heihe river basin and presented the regional distribution of net radiation over different eco-hydrological land surfaces. I don’t think the net radiation is a big problem for the surface energy balance, compared to the other components, like sensible heat, latent heat and ground heat fluxes. Of course, the surface energy balance system model can be used to estimate the net radiation, but it would be more interesting to show the results for the other components of the surface energy balance. Please see below for my comments:

  1. The title is misleading. Surface energy balance usually refers to the balance between turbulent heat fluxes (latent and sensible heat fluxes) and the available energy (net radiation minus ground heat flux) at the land-atmosphere interface. I would suggest remove the ‘surface energy balance’ from the title.
  2. How is the regional distribution of net radiation evaluated or validated as there are no ground-based measurements available?
  3. The remote sensing data and model description is not well-organized. It might be better to separate it into two parts.
  4. Please specify theta_0 and theta_a in equation 11.
  5. Figure 2: Please check the plots of soil temperature for midstream and downstream regions. Do these two regions have the same soil temperature?
  6. Line 250: ‘Figure 2’ should be ‘Figure 3’.
  7. Figure 3: The measured net radiation is at site-level, is the estimated net radiation also for these three sites?

Author Response

(The authors gave the same response as above.)

Round 2

Reviewer 1 Report

Accepted.

Author Response

Dear Reviewer,

Thank you very much for your acceptance to the responses regarding your comments in round 1 concerning our manuscript (atmosphere-964561). These comments were valuable and very much helpful for improving our paper. In round 2, we have revised the conclusions section in the revised manuscript as per your suggestion (Please see in the attachment). The changed and added portion have been highlighted in the revised manuscript with the changes tracked.

Best Regards.

Sincerely yours,

Dr. Md Masudur Rahman

Dr. Wanchang Zhang

Arfan Arshad

Reviewer 3 Report

Review of the revision of the manuscript "Regional Distribution of Net Radiation Flux over Different Ecohydrological Land Surfaces"

1. Line 85, change 'evaluate' to 'examine'. I think three point measurements of Rn is not enough to evaluate a regional distribution of Rn.

2. Please check the SEBS model, I don't think that the net radiation is computed based on the similarity relationship. 

3. From equation 11, the downward solar radiation and albedo are two most important parameters. Can you show some comparison and validation of these two variables between the estimation and measurements, which may explain partially the uncertainties in Rn over the three sites?

4. In Figure 3, the measured Rn is compared with the average estimated Rn over the three regions. Can you compare the measured Rn with estimated Rn only around the three sites, not regional averages of estimated Rn?  Also, please use the same ranges for y-axis and x-axis. I thought the estimated Rn were generally overestimated at a glance.

5. Please change 'net radiation flux' to 'net radiation'.

Author Response

Dear Reviewer,

Thank you very much for your acceptance to the responses regarding your comments in round 1 concerning our manuscript (atmosphere-964561). These comments were valuable and very much helpful for improving our paper. Again, thank you very much for your comments in round 2 concerning our manuscript (atmosphere-964561). These comments are also valuable and very much helpful for improving our paper. We have studied your comments carefully and have revised our manuscript accordingly, which we hope meet with your approval. The changed and added portion have been highlighted in the revised manuscript with the changes tracked.

Please see the attachment for the point by point responses.

Best Regards.

Sincerely yours,

Dr. Md Masudur Rahman

Dr. Wanchang Zhang

Arfan Arshad
